# A Nasal Vaccine Candidate, Containing Three Antigenic Regions from SARS-CoV-2, to Induce a Broader Response

**DOI:** 10.3390/vaccines12060588

**Published:** 2024-05-28

**Authors:** Yadira Lobaina, Rong Chen, Edith Suzarte, Panchao Ai, Alexis Musacchio, Yaqin Lan, Glay Chinea, Changyuan Tan, Ricardo Silva, Gerardo Guillen, Ke Yang, Wen Li, Yasser Perera, Lisset Hermida

**Affiliations:** 1Research Department, China-Cuba Biotechnology Joint Innovation Center (CCBJIC), Lengshuitan District, Yongzhou 425000, China; ylobainamato@ccbjic.com (Y.L.); doris@ccbjic.com (R.C.); aipanchao@ccbjic.com (P.A.); alexis.musacchio@cigb.edu.cu (A.M.); ava@ccbjic.com (Y.L.); trim@ccbjic.com (C.T.); young@ccbjic.com (K.Y.); liwen@ccbjic.com (W.L.); 2R&D Department, Yongzhou Zhong Gu Biotechnology Co., Ltd., Yangjiaqiao Street, Lengshuitan District, Yongzhou 425000, China; 3Yongzhou Development and Construction Investment Co., Ltd. (YDCI), Changfeng Industry Park, Yongzhou Economic and Technological Development Zone, No. 1 Liebao Road, Lengshuitan District, Yongzhou 425000, China; 4Research Department, Center for Genetic Engineering and Biotechnology, Havana 10600, Cuba; edith.suzarte@cigb.edu.cu (E.S.); glay.chinea@cigb.edu.cu (G.C.); gerardo.guillen@cigb.edu.cu (G.G.); 5Science and Innovation Directorate, BioCubaFarma, Independence Avenue, No. 8126, Corner 100 Street, Havana 10800, Cuba; rsilva@oc.biocubafarma.cu

**Keywords:** S2 fibre, nucleocapsid, RBD, SARS-CoV-2, chimeric protein, intranasal vaccine

## Abstract

A chimeric protein, formed by two fragments of the conserved nucleocapsid (N) and S2 proteins from SARS-CoV-2, was obtained as a recombinant construct in *Escherichia coli*. The N fragment belongs to the C-terminal domain whereas the S2 fragment spans the fibre structure in the post-fusion conformation of the spike protein. The resultant protein, named S2NDH, was able to form spherical particles of 10 nm, which forms aggregates upon mixture with the CpG ODN-39M. Both preparations were recognized by positive COVID-19 human sera. The S2NDH + ODN-39M formulation administered by the intranasal route resulted highly immunogenic in Balb/c mice. It induced cross-reactive anti-N humoral immunity in both sera and bronchoalveolar fluids, under a Th1 pattern. The cell-mediated immunity (CMI) was also broad, with positive response even against the N protein of SARS-CoV-1. However, neither neutralizing antibodies (NAb) nor CMI against the S2 region were obtained. As alternative, the RBD protein was included in the formulation as inducer of NAb. Upon evaluation in mice by the intranasal route, a clear adjuvant effect was detected for the S2NDH + ODN-39M preparation over RBD. High levels of NAb were induced against SARS-CoV-2 and SARS-CoV-1. The bivalent formulation S2NDH + ODN-39M + RBD, administered by the intranasal route, constitutes an attractive proposal as booster vaccine of sarbecovirus scope.

## 1. Introduction

Over the past twenty years, three novel and pathogenic coronaviruses have caused human epidemics or pandemics. In 2003, the severe acute respiratory syndrome coronavirus (SARS-CoV) exhibited a 10% case–fatality ratio (CFR) [1]. In 2012, the Middle East respiratory syndrome coronavirus (MERS-CoV) emerged, with 35% CFR [1]. The last one, SARS-CoV-2, appeared in 2019 with the lowest CFR, but its high transmissibility provoked more than 650 million reported COVID-19 cases and more than 6.6 million deaths worldwide [2].

For SARS-CoV and MERS-CoV, bats are likely the original reservoir. They later adapted to palm civets (SARS-CoV) and dromedary camels (MERSCoV) [3]. The source of SARS-CoV-2 has yet to be definitively determined; however, bats, linked to other animal hosts as intermediate, are the most likely possibility [4]. Bats are a natural reservoir of many coronaviruses, and it is solidly documented, particularly for sarbecoviruses [5]. Based on the previous elements, along with the fact that coronaviruses can evolve rapidly, it is expected that other pathogenic coronaviruses will emerge in the future [3]. In addition, the widespread and persistent circulation of SARS-CoV-2 suggests the possibility of recombination events with other coronaviruses, which could originate new threatening viruses with unknown characteristics [6].

Such persistence circulation of SARS-CoV-2 is due to the limited breadth of the immunity and the lack of durable mucosal immunity generated by both the current COVID-19 vaccines and natural infections [7,8]. Mucosal immunity is recognized as critical to halt the virus transmission [9]; its proper stimulation may prevent viral entry into mucosal cells, which will prevent infection and decrease the potential for asymptomatic transmission of different coronaviruses [10].

Based on the real threat of upcoming zoonosis events related to coronaviruses, as well as the limitations of the current vaccines, it is crucial to develop a next generation of vaccines able to provide broader protection, and at the same time, induce a broad mucosal immunity.

For developing universal coronavirus vaccines, two main strategies are being employed: multiple antigens based on spike (S) proteins (multivalent) and conserved antigens or regions among coronaviruses. A multivalent approach relies on the inclusion of many highly immunogenic regions, and subsequently, the generated immunity is directed to the variants contained within the vaccine design [10,11,12,13]. On the other hand, the approach based on conserved antigens could reach a broad scope of protection in line with the level of conservation of the proteins included.

Two of the most relevant regions of the virus to be included in a pancorona vaccine are the nucleocapsid (N) protein and the S2 subunit of the spike protein [14,15]. The N protein, a highly conserved protein among coronaviruses, has been identified as target of cell-mediated immunity (CMI) in humans which correlates with protection [16]. Accordingly, Matchett et al., 2021 reported that the N protein, obtained in a recombinant viral vector, is able to protect mice against SARS-CoV-2 challenge [17]. Dangi et al., 2021 also demonstrated that immunization with a preparation containing N and spike proteins improved distal protection in mouse brains compared to the administration of the spike protein alone [18]. On the other hand, S2 is the most conserved region in the S protein; it is involved in the fusion process for the virus entrance and contains T cell epitopes in humans [19]. One of these T cell epitopes is conserved among several coronaviruses, which might be associated with the rapid response upon SARS-CoV-2 infection and, consequently, the decrease of severity in humans [19]. Within the S2 subunit, the region corresponding to the fibre structure in the post-fusion conformation is an interesting target for vaccine design. This region contains the conserved CMI epitope and, under a proper conformation, would be able to induce functional Abs with different effector mechanisms [20].

In the present study, the two abovementioned elements associated with a broader scope second generation vaccine design are addressed: (1) candidates based on conserved regions from SARS-CoV-2 virus, and (2) induction of mucosal immunity. In addition, the subunit platform of vaccines is selected due to its proven safety profile [21].

A novel chimeric protein was designed based on two selected conserved regions of N and S2 proteins from SARS-CoV-2. The N fragment belongs to the C-terminal domain (CTD), whereas the S2 fragment spans the fibre structure in the post-fusion conformation of the S protein. The resultant chimeric protein, named S2NDH, was obtained in *E. coli* with more than 95% purity and characterized. Its immunogenicity was evaluated in mice by the intranasal and subcutaneous routes. Moreover, a nasal bivalent formulation, containing S2NDH + RBD protein and the CpG oligodeoxynucleotide (ODN) ODN-39M as adjuvant, was also evaluated. A cross-reactive, humoral and cell-mediated, immunity against the N protein; in addition to neutralizing Ab against SARS-CoV-2 and SARS-CoV-1, were generated by the bivalent formulation. Interestingly, such a response was obtained in both systemic and mucosal compartments under the typical Th1 pattern.

## 2. Materials and Methods

### 2.1. Biological Reagents

The pET28a (+) plasmid vector and *Escherichia coli* BL21 (DE3) cell strain (Sangon Biotech, Shanghai, China): F– ompT gal dcm lonhsdSB(rB- mB-) k(DE3 [lacI lacUV5-T7 gene 1 ind1 sam7 nin5]) were used for expressing the chimeric gene [22].

The recombinant antigens were purchased from Sinobiological Inc. (Beijing, China). Nucleocapsid proteins were from SARS-CoV-2, Delta (40588-V07E29) and Omicron (40588-V07E34) variants and from SARS-CoV-1 (40143-V08B). The S2 protein was from SARS-CoV-2 Ancestral strain (40590-V08H1). The RBD protein was from the SARS-CoV-2 Delta (40592-V08H90) and Omicron (40592-V08H123) variants and SARS-CoV-1 (40150-V08B2).

The peptide N_351–365_ from SARS-CoV-2 (ILLNKHIDAYKTFPP) was synthesized with ≥97% purity by Zhejiang Peptides Biotech (China).

The ODN-39M, a 39 mer, whole-phosphodiester backbone oligodeoxynucleotide (5′-ATC GAC TCT CGA GCG TTC TCG GGG GAC GAT CGT CGG GGG-3′), was synthesized by Sangon Biotech (China).

Polyclonal anti-N antibody was purchased from Sinobiological (Cat No. 40588-T62).

Human sera from COVID-19 convalescent (N = 10) and negative (N = 10) individuals were obtained by the team from Guangdong Eighth People’s Hospital. Samples were collected as part of a study approved by the Institutional Ethics Committee from the Eighth People’s Hospital of Dongguan (Guangdong Province, China) as previously reported [23]. Informed written consent was obtained from each participant before enrolment in the study.

### 2.2. Cloning and Expression of S2NDH

The chimeric gene composed of the DNA sequences coding for the S2NDH protein was chemically synthesized following a design that contained the S2 fragment (800–1020) + linker GGSSGG + N fragment (248–371) + linker GGSSGG + (His)6. The chimeric gene was amplified by PCR using the corresponding primers. The amplified band was purified and cloned into pGEM-T Easy Vector (Promega, USA). Positive clones were tested by restriction analysis, and the chimeric genes were fully sequenced (Sangon Biotech, China). The fragment encoding for the chimeric protein was then cloned into pET28a plasmid (Sangon Biotech, China). Positive clones were identified by restriction analysis and conserved under the name pS2NDH.

The *E. coli* strain BL21 (DE3) (Sangon Biotech, China) was transformed with the recombinant plasmid by electroporation. The clone selected was later inoculated in LB medium containing 30 µg/mL kanamycin, and cultured at 37 °C until reaching an OD_600_ of 0.6–0.8. IPTG was added at a final concentration of 0.5 mM and then continued culturing for 4 h at 37 °C. The final culture was centrifuged at 10,000× *g* for 15 min at 4 °C, and whole cells were collected to be analysed by SDS-PAGE and Western blot.

### 2.3. Purification of S2NDH

The cell clone exhibiting the best expression level was amplified in 5 L culture of LB medium (1% glucose, 30 µg/mL kanamycin) at 37 °C. When the OD_600_ of the culture reached values of 0.6–0.8, IPTG was added at a final concentration of 0.5 mM, and then continue culturing for 4 h at 37 °C.

After centrifugation at 10,000× *g* for 15 min at 4 °C, the resulting biomass was harvested. For cell disruption, cells were resuspended in buffer PBS, pH 7.4. They were subsequently disrupted at 4 °C in a sonicator (JY88-II, Tianjin Jinli Instrument and Equipment Technology Development Co., Ltd., Tianjin, China). Then, the sample was centrifuged at 10,000× *g* for 15 min at 4 °C, and the pellet was dissolved in 8 M urea and 50 mM Tris-HCl, pH 8.0 denaturing buffer. Another centrifugation cycle was carried out at 10,000× *g* for 15 min at 4 °C, and the supernatant was collected for the following step of ion exchange chromatography.

The SP Sepharose fast flow ion exchanger was selected. The matrix was equilibrated with 8 M urea and 50 mM Tris-HCl, pH 8.0 buffer. The elution was performed with the same buffer but using 300 mM NaCl. The collected sample was applied to gel filtration chromatography. The matrix Superdex 75 was used, and the running was performed in 20 mM Tris and 8 M urea, pH 8.0. The collected samples, containing the protein S2NDH, were mixed and subsequently subjected to a third chromatographic step, immobilized metal affinity chromatography (IMAC). A Ni-NTA matrix was equilibrated with 8 M urea, 50 mM Tris-HCl, and 50 mM imidazole, pH 8.0. The S2NDH protein was eluted using 500 mM imidazole in the same buffer.

Finally, the protein was subjected to a renaturing process by dilution in the ratio of 1:100 using buffer 110 mM Tris and 1.1 M Gua-HCl, pH 8.0. After slow stirring at room temperature for 24 h, a filtration step was introduced to remove impurities and the purified protein was transferred to the final buffer: 10 mM Tris, 150 mM NaCl, 6 mM EDTA, pH 6.9. The resultant preparation was concentrated reaching a final concentration of 1 mg/mL.

### 2.4. In Vitro Incubation of S2NDH with ODN-39M

The chimeric protein S2NDH was incubated with the ODN-39M, as previously described [24], with few modifications. Briefly, in a 100 µL reaction, 40 µg of S2NDH were mixed with 60 µg of ODN-39M in buffer 10 mM Tris, 6 mM EDTA, pH 6.9. The resulting preparations were incubated for 30 min at room temperature (RT) (20–25 °C) and then stored at 4 °C until immunogen formulation.

### 2.5. Protein Analysis

The samples protein concentration was determined by BCA assay (Pierce Thermo Scientific, Rockford, IL, USA). Protein samples were analysed using 10–15% sodium dodecyl sulphate polyacrylamide gel electrophoresis (SDS-PAGE) [25]. SDS-PAGE gels were stained with Coomassie blue and scanned (iBright 1500, Invitrogen, Waltham, MA, USA). The percentage corresponding to the protein band of interest was estimated by densitometry analysis using ImageJ (version 1.41) software.

For immune identification, Western blot assay was carried out. Briefly, protein samples were electro-transferred from acrylamide gels to Immobilon-P membranes (Merck-Millipore, Dublin, Ireland), as described in [26]. The membrane was blocked with 5% skim milk in phosphate-buffered saline (PBS) for 1 h at RT. Then it was washed three times in PBS–0.05% Tween 20 solution (PBS-T) and allowed to react with the anti-SARS-CoV-2 Nucleocapsid polyclonal Ab generated in rabbit (Sinobiologicals, Beijing, China), at 1:2500 dilution, or with the mouse Mab anti-His Tag—peroxidase conjugated (Sinobiological, Beijing, China) at 0.2 µg/mL, for 1 h at RT. In the first case, after washing, the membrane was incubated with peroxidase-conjugated goat anti-rabbit IgG (Chemicon, Rolling Meadows, IL, USA) at a 1/300 dilution for 1 h at RT. Afterwards, the membrane was washed again, and the antigen-antibody reaction was revealed by incubation with aminoethyl carbazole (AEC) substrate solution (0.2 mg/mL 3-amino-9-ethylcarbazole and 0.03% H_2_O_2_ in 50 mM NaAc solution) at RT until the color was developed.

### 2.6. Transmission Electron Microscopy

For microscopy analysis, a specialized service was contracted with ETest company (Changsha, China). Briefly, samples of the purified chimeric protein S2NDH, alone or mixed with ODN-39M, were placed on a freshly glow-discharged, 400-mesh copper grid coated with formvar and carbon. After 2 min of sample absorption, the grids were washed with water and uranyl acetate stain was applied. Following 4 min of staining, grids were wick-dried using Whatman no. 1 filter paper and allowed to air-dry for 20 min.

A Transmission Electron Microscope HT 7800 (Hitachi, Tokyo, Japan) was employed for sample visualization. An acceleration voltage of 120 Kv and three magnifications, 25,000×, 50,000× and 100,000×, were used. Eight random fields per sample were photographed and analysed. The average particle size was estimated by digital measurement of the particle diameter using Image J software (version 1.4.3, Bethesda, MD, USA).

### 2.7. Antigenic Characterization Using SARS-CoV-2 Human Sera

Ninety-six-well high-binding polystyrene plates (Costar Corning, Kennebunk, ME, USA) were coated with 3 μg/mL of S2NDH (alone or mixed with ODN-39M), and ODN-39M as control, in sodium carbonate-sodium bicarbonate buffer, and incubated overnight at 4 °C. Then, the plates were blocked with 5% skim milk (Oxoid, Basingstoke, UK) 1 h at 37 °C. After five washes with PBS-T, 100 μL of the serum sample diluted in 1% milk –PBS-T was added. The samples were incubated for 2 h at 37 °C. After being washed five times with PBS-T, conjugated horseradish peroxidase–goat anti-human IgG [1:20,000] (Sigma-Aldrich Co., Schnelldorf, Germany) was added for a period of 1 h at 37 °C. Afterwards, another cycle of washes was carried out, and 100 μL of OPD substrate solution (Sigma-Aldrich Co., Germany) was added. The plates were incubated for 10 min at RT, and finally the reaction was stopped by adding 0.2N sulphuric acid. Optical density (O.D) at 492 nm was measured (FilterMax F3, Molecular Devices, San Jose, CA, USA). The data are represented as O.D measures.

### 2.8. Immunization Experiments

Six-to-eight-week-old female Balb/c mice (inbred, H-2d) were employed (Beijing Vital River Laboratory Animal Technology Co., Ltd., Beijing, China). The standard of laboratory animal room complied with the national standard of the people’s Republic of China GB14925-2010. The immunization protocols were approved by Institutional Animal Care and Use Committee.

Some immunogens for subcutaneous (sc) administration contained aluminum hydroxide (alum; Alhydrogel, Invitrogen), as adjuvant, at a final concentration of 1.4 mg/mL. All immunogens were dissolved in sterile PBS, and a dose of 10 μg of protein was used. For intranasal (in) and subcutaneous (sc) administrations, a volume of 50 μL and 100 μL was employed, respectively.

In the first mouse experiment, animals were distributed into seven groups of five animals each and immunized with three doses, administered on days 0, 7, and 21. The S2NDH protein was administered by the intranasal route alone (S2NDH in) or mixed with ODN-39M (S2NDH + ODN in). On the other hand, S2NDH was evaluated subcutaneously adjuvated in alum (S2NDH Al sc), mixed with ODN-39M (S2NDH + ODN sc) and using both adjuvants (S2NDH + ODN Al sc). As placebo controls, a group receiving alum by the sc route (PBS Al sc) and another receiving PBS intranasally (PBS in) were included.

In the second mouse experiment, groups of six animals were intranasally immunized with three doses, administered on days 0, 15 and 30. In this case, S2NDH was evaluated alone, mixed with ODN-39M, and in a bivalent formulation containing ODN-39M and RBD protein from SARS-CoV-2 Delta strain (S2NDH + ODN + RBD in). As controls, RBD alone (RBD in) and PBS were administered.

At the time points of 19 and 27 days after the last dose, for the first and second experiment, respectively, different kind of samples were collected: sera, bronchoalveolar fluid, (BALF) and spleens.

### 2.9. Evaluation of Humoral Immune Response by ELISA

The systemic and mucosal antibody response was evaluated by ELISA. Anti-IgG, subclasses, and -IgA ELISAs were conducted as previously described [27]. Briefly, 3 µg/mL of recombinant protein was used to coat 96-well high-binding plates (Costar, USA). Plates were subsequently blocked with 2% skim milk solution. Samples were added in duplicates, starting from 1:100 dilution in the case of sera, whereas BALF were directly assayed. Specific horseradish peroxidase conjugates (Sigma, Livonia, MI, USA) were used. As a substrate, OPD (Sigma, USA)/hydrogen peroxide solution was employed. Plates were incubated for 10 min in the dark, and the reaction was stopped with 2 N sulphuric acid. The O.D was read at 492 nm in a multiplate reader (FilterMax F3, Molecular Devices, USA). For the evaluations in serum samples the data was represented, mainly, as log10 titers. The arbitrary units of titers were calculated by plotting the O.D values obtained for each sample in a standard curve (hyper-immune serum of known titer). The positivity cut-off was established as 2 times the average of O.D obtained for a pre-immune sera pool. Otherwise, and also for the antibodies detected in BALFs, data were represented as O.D at 492 nm.

### 2.10. IFN-γ ELISPOT

IFN-γ ELISPOT assay was carried out using a specific antibody pair developed for the detection of this cytokine in mice samples (Mabtech, Nacka Strand, Sweden). Spleen cells were isolated in RPMI culture medium (Gibco, Thermo Fisher Scientific Suzhou, China). Samples (five mice per group) were singly processed, with the exception of the placebo group, for which a pool of three randomly selected mice was evaluated. Duplicate cultures (5 × 10^5^ and 1 × 10^5^ splenocytes per well) were incubated for 48 h at 37 °C and 5% CO_2_ in a 96-well round-bottom plate (Costar, USA) with a final concentration of 10 µg/mL of each stimulating agent: N_351-365_ peptide, N or RBD proteins, concanavalin A (ConA) (Sigma, USA), or medium. The content of the plate was then transferred to an ELISPOT pre-coated plate and incubated for 16–20 h at 37 °C and 5% CO_2_. The successive steps were carried out following the manufacturer’s recommendations. Finally, the spots were counted using a stereoscopic microscope (AmScope SM-1TSZ, Irvine, CA, USA) coupled to a digital camera.

### 2.11. Pseudotyped VSV-Based Neutralization Assay

SARS-CoV-2 S protein pseudotyped vesicular stomatitis virus (VSV)-based assay was employed to quantify the neutralizing capacity of the sera [28]. A commercial kit containing a viral stock of VSV pseudotyped with the whole S protein from the Ancestral strain and a Luciferase substrate solution were acquired (Darui Biotech, Guangzhou, China). In a 96-well culture plate (Costar, USA), sera dilutions were incubated with the recommended concentration of virus for 1 h at 37 °C and 5% CO_2_. Two columns of the plates were reserved for virus control (VC, without serum sample), and cell control (CC, without virus). Later, 2 × 10^4^ Huh-7 cells (cell line provided by Darui Biotech, China) were added to each well and incubated for 24 h at 37 °C and 5% CO_2_. After this period, 150 µL of supernatant from each well was removed, and 100 µL of Luciferase substrate solution was added. The plates were incubated for 2 min in the dark, and then the content of each well was resuspended and transferred to a white opaque 96-well plate (Costar, USA). The luminescence was read using a FilterMax F3 microplate reader (Molecular Device, USA). The calculation of the inhibition percentage was done as follows:Inhibition rate = (1 − (average of luminescence for sample − average luminescence for CC)/(average luminescence for VC − average luminescence for CC)) × 100%

Positive and negative sera controls were included in the assay. The positive control consists of a pool of mice sera with a known neutralizing titer 1:4000. The negative control comprises a pool from the corresponding placebo group. The assay fulfilled with the quality criteria recommended for this kind of test.

To determine the neutralizing antibody (NAb) titers (EC_50_) vs. SARS-CoV-1 in sera from the bivalent formulation immunized group, a professional service was contracted to Darui Biotech, China. In this case, the pseudotyped VSV system, carrying the spike protein from SARS-CoV-1 was used. As positive control, a serum with known neutralizing titer established by CRO was used. The negative control consisted of a pool from the placebo group. In all cases the EC_50_ was calculated using the Reed–Muench method.

### 2.12. Statistical Analysis

Graph Pad Prism version 5.00 software (Graph-Pad Software, San Diego, CA, USA) was employed for the statistical analysis. The Ab titer values were transformed to log10 to adjust the data to a normal distribution. For the non sero-converting sera, an arbitrary titer of 1:50 was considered. One-way ANOVA was used as parametric tests for multiple group comparisons, followed by a Tukey’s post-test. In the particular cases that required it, non-parametric multiple comparisons using the Kruskal–Wallis test and Dunn’s post-test was employed. The statistical criteria followed the standard considerations for *p* values: ns, *p* > 0.05; *, *p* < 0.05; **, *p* < 0.01; ***, *p* < 0.001.

## 3. Results

### 3.1. Cloning, Expression, and Purification of S2NDH Construct

The DNA sequence coding for the chimeric protein S2NDH (formed by the fragment S 800–1020 fused to the fragment N 248–371, both based on SARS-CoV-2 Delta variant) was cloned into the pET-28 vector. *E. coli* BL21 was then transformed with the plasmid containing the recombinant chimeric construct and grown in LB medium. As shown in Figure 1a, a reinforced band of MW around 40 kDa, with the theoretical size of the S2NDH protein, was visualized. It accounted for 15% of the total cellular proteins associated with the pellet fraction upon disruption. In addition, the identity of the band was corroborated by specific immune recognition with anti-His tag Ab (Figure 1b).

After cellular disruption, the protein S2NDH associated with the pellet fraction was extracted with 8M urea and applied into the SP Sepharose fast-flow matrix. The chimeric protein eluted at 300 mM NaCl was later subjected to gel filtration chromatography. The collected fractions were finally applied to the IMAC matrix. After elution in a high concentration of imidazole, the protein sample was renatured and characterized by SDS-PAGE. As shown in Figure 1c, a highly pure final preparation of S2NDH protein was obtained, with an estimated purity level of more than 95%.

The purified sample of the chimeric protein S2NDH was examined by TEM. Interestingly, particles with spherical morphology and a nanometric size of ~10 nm were observed (Figure 1d).

Since the ODN-39M is the mucosal adjuvant selected to evaluate the chimeric protein as vaccine candidate, and it has the ability to interact with the N protein of SARS-CoV-2 [24], the combination S2NDH + ODN-39M was also examined by TEM. As shown in Figure 1e, large aggregates structures were visualized in this preparation compared to the protein alone, indicating some level of interaction between both molecules.

The recognition of the chimeric protein S2NDH, as well as the mixture S2NDH + ODN-39M, was evaluated by ELISA using human sera from COVID-19 convalescent and negative donors which were collected during an outbreak of SARS-CoV-2 Omicron variant in 2022 at Guangdong province, China. As shown in Figure 1f, both preparations were recognized by the human sera positive to SARS-CoV-2, without statistical significant differences between them (*p* > 0.05).

### 3.2. Immunological Evaluation of S2NDH in Balb/c Mice

The immunogenicity of S2NDH protein was studied in Balb/c mice with two different adjuvants, ODN-39M and/or alum, and administered by the intranasal or subcutaneous routes (Figure 2a).

The IgG response generated in sera against S2NDH, S2 (from SARS-CoV-2 Ancestral strain), and N (from SARS-CoV-2 Delta strain) recombinant proteins was evaluated by ELISA. All groups immunized with the chimeric protein showed high levels of anti-S2NDH and anti-N antibodies regardless the formulation administered or the inoculation route used (*p* > 0.05) (Figure 2b,c). However, the antibody response against S2 protein detected was mostly negative in all groups; only few mice showed seroconversion (Figure 2d).

In turn, when the IgG subclasses were evaluated against the N protein, the highest levels of IgG2a were obtained for the groups receiving S2NDH with ODN-39M by the intranasal or subcutaneous route, showing statistically significant differences compared to the group receiving the chimeric protein with alum by sc route (*p* < 0.05) (Figure 3a). Conversely, the IgG1 levels were significant lower for the two intranasal immunized groups (*p* < 0.01) (Figure 3b).

The IgA antibodies against the N protein were evaluated in BALF (Figure 3c). As expected, only the two groups intranasally immunized with S2NDH showed a positive response, without statistical differences between them (*p* > 0.05).

The VSV pseudotyped virus system, carrying the spike protein of the heterologous strain (Ancestral variant), was used to detect the neutralizing activity in sera. As a result, no neutralizing activity was obtained.

On the other hand, an ELISPOT assay was carried out to detect IFN-γ secreting spleen cells upon in vitro stimulation with the N_351–365_ conserved peptide. As shown in Figure 3d, only the group intranasally inoculated with S2NDH + ODN-39M exhibited 100% response, with a clear trend to develop a higher IFN-γ secretion level.

### 3.3. Assessment of the Cross-Reactive Anti-N Immunity Induced after Intranasal Administration of S2NDH + ODN-39M

The cross-reactivity of both arms of the immune response generated against the N protein in mice intranasally immunized with the formulation S2NDH + ODN-39M was also explored. 

Samples from the group receiving S2NDH + ODN-39M (in) were analysed against heterologous N antigens. As shown in Figure 4a, high levels of IgG Abs against N proteins from SARS-CoV-2 Omicron and SARS-CoV-1 were obtained in sera. A similar behaviour was detected for the anti-N IgA response measured in BALFs (Figure 4b).

In turn, the IFN-γ ELISPOT assay was also carried out using N proteins from SARS-CoV-2 Delta and Omicron and from SARS-CoV-1 as stimulating agents (Figure 4c). Interestingly, a positive response was observed for the three antigens assayed, indicating the induction of a broad scope cellular immunity after intranasal immunization with the S2NDH + ODN-39M preparation.

### 3.4. Immunological Evaluation of S2NDH + ODN-39M + RBD by the Intranasal Route

Considering that S2NDH monovalent preparations were not able to induce a neutralizing antibody response, the RBD protein was added to the formulation as inducer of functional NAbs. The resulting bivalent formulation, S2NDH + ODN-39M + RBD, was also evaluated in Balb/c mice by the intranasal route.

In sera, the bivalent formulation induced high levels of anti-N (Figure 5a) and anti-RBD IgG Abs (Figure 5b). Interestingly, a clear adjuvant effect of the S2NDH + ODN-39M preparation over the RBD was detected, since RBD alone, administered by the intranasal route, was poorly immunogenic (*p* < 0.001).

The IgG subclass pattern against RBD was also explored. Interestingly, higher titers of IgG2a anti-RBD were induced by the bivalent formulation compared with RBD alone (*p* < 0.001) (Appendix A), supporting the development of a Th1 pattern.

The mucosal immunity, measured as IgA levels against each antigen, was evaluated in BALFs. High and similar levels of anti-N IgA Abs were induced by the S2NDH + ODN-39M and the bivalent formulation S2NDH + ODN-39M + RBD (*p* > 0.05) (Figure 5c). In turn, according to the anti-RBD IgG Abs in sera, significant levels of anti-RBD IgA Abs were detected in the group receiving the bivalent formulation (*p* < 0.001) (Figure 5d). This result evidences the adjuvant effect of the S2NDH + ODN-39M preparation over RBD mucosal immunity.

The CMI response, measured by IFN-γ ELISPOT assay after in vitro stimulation with the conserved peptide N_351–365_ and RBD, was also carried out (Figure 5e and Figure 5f, respectively). According to the humoral immune response, animals receiving the bivalent formulation by the intranasal route exhibited a positive response against both stimuli. Of note, no response was detected for the group intranasally immunized with RBD alone, also supporting the adjuvant effect of the S2NDH + ODN-39M over the anti-RBD CMI.

Furthermore, the cross-reactivity of the humoral immune response elicited by the intranasal administration of the bivalent formulation was also assessed. As expected, considering the previous results for the S2NDH + ODN preparation, the anti-N antibody response showed cross reactivity with N proteins from SARS-CoV-2 Omicron variant and SARS-CoV-1 viruses in sera (Figure 6a) and BALF (Figure 6b). Interestingly, the anti-RBD antibody response in sera and BALF (Figure 6c and Figure 6d, respectively) also showed cross-reaction with the RBD protein from SARS-CoV-2 Omicron variant and with RBD from SARS-CoV-1.

### 3.5. Neutralizing Capacity of Antibody Response Elicited by S2NDH + ODN-39M + RBD

The VSV pseudotyped virus system, carrying the spike protein of the SARS-CoV-2 heterologous strain (Ancestral variant), was used to detect the neutralizing capacity of sera from the group intranasally immunized with S2NDH + ODN-39M + RBD. As a result, the 100% of sera exhibited neutralizing Ab titers (EC50 > 1:700), with a mean titer value of 3043 (Figure 7a, left panel). In addition, when the same serum samples were tested with the VSV pseudotyped with S protein from SARS-CoV-1, again, the 100% of sera exhibited a positive neutralizing response with a mean value of EC50 of 131 (Figure 7a, right panel).

Furthermore, the neutralizing Ab capacity was also evaluated in BALF samples (Figure 7b); 100% of samples were able to neutralize the VSV pseudotyped with S protein from SARS-CoV-2, Ancestral strain.

## 4. Discussion

Nowadays, there is an increasing concern about the long-term safety of the mRNA vaccines and those based on adenovirus vectors [29]. Despite the proven capacity of these new platforms to induce high immunogenicity levels, their long-term effect has not been previously documented. As an alternative, the present study proposes vaccine candidates based on the subunit platform, which combines one chimeric antigen obtained from *E. coli* (as inducer of CMI) and RBD, obtained also as a recombinant protein, but in mammalian cells, mainly as an inducer of neutralizing Abs. In particular, the bioinformatic design of the chimeric protein S2NDH is novel, since in the same molecule, conserved regions (N CTD and the S2 fibre) of two different structural proteins of the SARS-CoV-2 virus, are present. The selected fragments have the potential capacity to be obtained in bacteria as independent structural domains bounded by a linking motif. Accordingly, the resultant chimeric construct was expressed at high levels in *E. coli*. It allowed to develop a simple and potentially scalable downstream process which produced protein preparations of high levels of purity. It was interesting that the chimeric protein S2NDH was able to form particles of around 10 nm. Probably, the presence of the N CTD region favoured such structures, as the capacity of the N protein to self-aggregate has been previously reported [24]. On the other hand, the addition of the ODN-39M as adjuvant in the vaccine preparation induced the formation of higher aggregated structures. This modification, upon ODN-39M addition, suggests an electrostatic interaction between both molecules. In fact, it is proposed that the N CTD region from coronaviruses is capable of interacting with the ARN [30].

For developing subunit nasal vaccines, despite having aggregated structures which are known to be immunogenic, it is crucial to define the vehicle to enhance such immunogenicity and consequently overcome the mucosal barrier. In the present study, the ODN-39M was selected based on its proven adjuvant capacity over viral proteins by parenteral routes [31,32,33]. Moreover, for nasal immunity enhancement, we previously demonstrated its capacity using the whole nucleocapsid protein as an antigen [34].

In the first mouse experiment, the immunogenicity of the chimeric protein measured by anti-N and anti-S2NDH IgG in sera, regardless of the adjuvant and route employed, was clearly evidenced. Such results are in contrast to those previously obtained by our group, in which mice immunized with the non-adjuvanted SARS-CoV-2 Nucleocapsid protein by the intranasal route did not elicit response in either sera or BALF [34]. In addition, 40% of animals receiving non-adjuvanted S2NDH intranasally were positive for CMI. This, along with the anti-N -IgA positive response in BALF and -IgG2a/IgG1 pattern in sera, points to a trend towards a Th1 response. Probably, the capsid-like particle (CLP) structure, evidenced by TEM for the S2NDH chimeric protein, favoured this kind of response. As expected, immunogenicity by the intranasal route was clearly enhanced by the addition of ODN-39M, which induced the formation of bigger aggregated structures, as was observed by TEM. This higher immunogenicity was clearly supported by the generated CMI response. Interestingly, 100% of responders were only obtained when S2NDH + ODN-39M was intranasally administered, despite high levels of anti-N IgG2a being obtained for this formulation by both the subcutaneous and intranasal routes. This strongly suggests that the intranasal route favoured the induction of CMI by S2NDH + ODN-39M. This fact can be explained due to mucosal tissues are enriched on plasmacytoid dendritic cells. It is well known that this kind of cell is activated by TLR9 agonists, as the CpGs ODNs [35], and the ODN-39M is per se a CpG ODN, with the distinction of possessing a whole phosphodiester backbone. This feature makes the formulation S2NDH + ODN-39M very attractive for preventive vaccination, since the thioate backbone has been identified as a factor contributing to the adverse reactions detected in therapeutics interventions [36].

Since the advantages of the intranasal route were proven, and considering also the relevance of mucosal response for respiratory infections, further experiments were focused on it. The cross-reactive profile of the humoral systemic and mucosal response, as well as the cellular immune response in spleen, induced by S2NDH + ODN-39M against the N protein, was deeply characterized. A broad response, covering until the N protein of SARS-CoV-1, was demonstrated. In particular, for CMI, the peptide N_351–365_ spanning a conserved motif among sarbecovirus which is immunodominant in Balb/c mice infected with SARS-CoV-2, was additionally used as stimulating agent. This peptide, presented in the Venezuelan equine encephalitis vector, partially protected mice from challenge with SARS-CoV-2 [37]; therefore, a positive CMI against it is relevant. In general, the scope of the CMI and humoral anti-N response induced by S2NDH + ODN-39M administered intranasally is in accordance with the homology percentages of SARS-CoV-2 N protein reported for different sarbecoviruses (87–99% of identity) [38].

The immune response against the S2 subunit was also measured. Unfortunately, no anti-S2 immunity was detected, either humoral or CMI. Concerning the humoral immunity, the lack of anti-S2 antibody response could be explained by differences in protein conformation between the commercial S2 protein used for the ELISAs and the S2 fragment containing in the chimeric protein, the latter corresponding to the fibre structure. It is widely reported that the native S2 protein suffers big conformational changes as part of the virus–host cell membrane fusion process during SARS-CoV-2 infection [39]. In particular, the S2 fibre structure remains hidden in the pre-fusion conformation, precisely due to its relevance as immune target this conformation is transient. This is in line with the low antibody response found in humans against the fusion peptide (aas 818–835) located in this fragment [40]. Furthermore, when the neutralizing Ab response was measured in sera from mice intranasally immunized with S2NDH + ODN-39M, no inhibition activity was detected, indicating that this kind of functional response might not being elicited. It is important to highlight that recent works point to the S2 region as a weaker inducer of neutralizing response. Authors demonstrated that other Ab effector functions such as Ab-dependent cell cytotoxicity (ADCC) and Ab-dependent cellular phagocytosis (ADCP), can be elicited by S2 and protects against relevant human coronaviruses [20,41]. Based on the measurements performed in the present study, we cannot rule out that some anti-S2 fibre response is being induced by the studied formulation, and even more to the point, if this potential response is functional, since no challenge experiment was carried out.

The S2 region also carries important T cell epitopes in humans. In particular, one of its conserved T cell epitopes might be associated with the rapid response upon SARS-CoV-2 infection [19]. However, our animal experiments were performed in Balb/c mice where, unlike the N protein, no inmunodominant T cell epitopes in the S2 region 800–1020 have been identified [37]. This fact could explain the lack of S2 (commercial protein)-specific IFN gamma secretion response observed in the spleen samples from mice immunized with S2NDH + ODN-39M formulation. Based on this element, we cannot conclude that the S2 region contained in the chimeric protein S2NDH will not be relevant in terms of CMI induction in humans.

One illustrative example of no detection of anti-S2 response after vaccination, but induction of functionality, was described by Maeda DLNF et al., 2021. The key conserved region of S2, the fusion peptide (FP), was inserted into the genome-reduced *Escherichia coli* to express the FP on the cell surface. Upon immunological evaluation in animals, neither Abs against FP nor CMI were induced after vaccination. Nevertheless, evidences of protection were achieved upon viral heterologous challenge, particularly in terms of symptomatology [42]. In our opinion, all the elements explained before support the inclusion of the S2 fibre region in the chimeric protein as a vaccine candidate.

On the other hand, the addition of the RBD protein to the nasal formulation, forming the bivalent formulation (S2NDH + ODN-39M + RBD), generated a more potent vaccine candidate, fulfilling the goal of inducing neutralizing Abs against SARS-CoV-2. Interestingly, the S2NDH + ODN-39M preparation exerted a potent adjuvant effect over RBD at the systemic and mucosal humoral immunity level, as well as at CMI level measured in the spleen, when administered by the intranasal route. The S2NDH + ODN-39M preparation can be considered an additional RBD immunity enhancer by the intranasal route, with the crucial contribution on favouring the anti-RBD CMI, since to our acknowledge, only few of them have demonstrated the potentiation of this arm of the immune system against RBD [43,44].

Another important advantage of this nasal bivalent formulation was the induction of antibody response against RBD of different variants of SARS-CoV-2 though the Ab levels against the RBD from the Delta variant, measured by ELISA, were higher than those detected against RBD Omicron. This pattern is in accordance to the high level of mutations accumulated in the spike protein and RBD region in the Omicron variant. The Omicron spike protein has at least 32 mutations from the original SARS-CoV-2, whereas the Delta variant has only 16. In turn, particularly for the RBD region, the Omicron variant has 13–15 amino acid mutations; 12 of them were new and only 3 are conserved with other variants [45].

Concerning the functionality of Abs induced, the bivalent candidate was able to elicit neutralizing Abs against pseudoviruses carrying both the spike protein from SARS-CoV-2 Ancestral variant and SARS-CoV-1. Several reports have demonstrated the cross-reactive profile of the immune response elicited by RBD, parenterally administered, against different strains of SARS-CoV-2 [46,47,48]; nevertheless, none of them provided data about neutralization against SARS-CoV-1. To our knowledge, this is the first report of inducing systemic anti-SARS-CoV-1 neutralizing Abs after nasal immunization with a formulation containing a recombinant RBD protein from SARS-CoV-2. Of note, the levels of neutralizing Abs against SARS-CoV-1 were lower than those detected against the SARS-CoV-2 Ancestral variant. Such differences can be explained by the low level of homology of the spike protein and RBD region between these two viruses.

The bivalent formulation described in the present work also induced anti-N cross-immunity until SARS-CoV-1 level, as was demonstrated for the S2NDH + ODN-39M component alone. In general, the broad immune response obtained with this nasal vaccine candidate is comparable to that elicited by the nasal vaccine candidate comprising a live-but-defective SARS-CoV-2 virus. The latter candidate cross-protected small animal models from sarbecovirus infection, such as SARS-CoV-1, SARS-CoV-2, and its variants [49]. In the case of the S2NDH + ODN-39M + RBD candidate, we recognize as a limitation of the present work the lack of protection assay. Such an experiment will be carried out after obtaining the immunogenicity results from previously infected and/or vaccinated animals, which indeed reflect the current immunological status of the population.

In general, for vaccine candidates based on subunit platforms, the induction of sarbecovirus neutralizing Abs have been intentionally targeted from the antigenic design. For instance, Cohen AA. et al., 2021 designed nanoparticles that presented 60 randomly arranged RBDs derived from the spike trimers of eight different sarbecoviruses. One of these constructs, mosaic-8, produced antisera that showed equivalent neutralization of SARS-CoV-2 variants, including Omicron, and protected from both SARS-CoV-2 and SARS-CoV-1 challenges in mice and nonhuman primates (NHPs) [11,50]. Similarly, Vishwanath S. et al., 2023 in silico designed immune-optimized and structurally engineered antigens, based on RBD fragments, which in the context of DNA platform, elicited broad humoral responses against SARS-CoV-1, SARS-CoV-2, WIV16 and RaTG13 in different animal models [51].

In the present study, our hypothesis is that RBD, combined with the S2NDH + ODN-39M preparation and presented by the intranasal route, favours the presentation of RBD cross-reactive epitopes and consequently, the generation of a broad neutralizing response until sarbecovirus level. In fact, several cross-neutralizing MAbs, targeting conserved regions in RBD, have been identified [52,53,54]. Another element to consider, which could partially contribute to the cross-neutralizing response observed, is that the RBD used in the present study comes from Delta variant of SARS-CoV-2, which could be more immunogenic than the Ancestral one. For instance, it has been reported that Gamma-adapted RBD vaccine is more immunogenic than the Ancestral RBD vaccine, including broader intra-strain neutralizing antibodies [55].

The cross-reactive profile of the mucosal and systemic humoral immunity against the N and RBD regions, along with the systemic anti-RBD CMI and the broad systemic anti-N CMI response, makes the S2NDH + ODN-39M + RBD bivalent formulation an attractive vaccine candidate for facing upcoming zoonosis events related to sarbecoviruses. In particular, the mucosal cross-reactive IgA Abs may contribute to stop the transmission of future zoonosis from the very beginning and accordingly, may early avoid a new pandemic. Of note, the cross-reactive immune response demonstrated in the present work was elicited in naive mice, therefore, in a population previously exposed to several SARS-CoV-2 variants and vaccines, it is expected a better outcome under the hypothesis of shared immunity enrichment, like the nature does through some subsequent viral infections [16,56].

## Figures and Tables

**Figure 1 vaccines-12-00588-f001:**
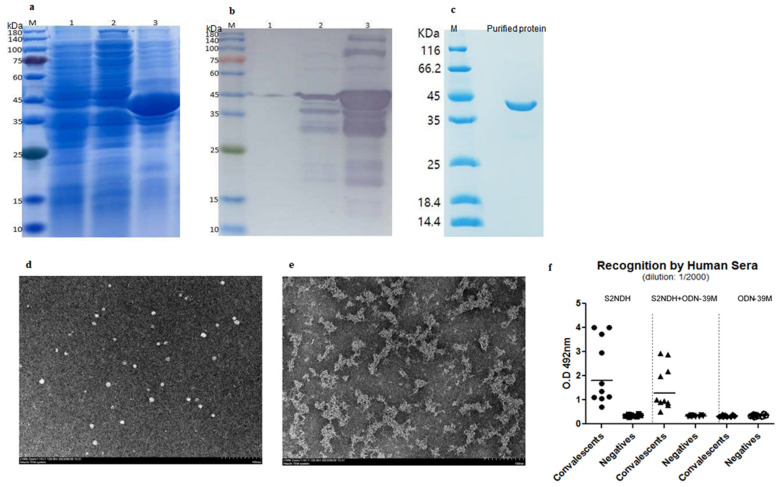
Obtaining, purification, and characterization of the chimeric protein S2NDH. Analysis of expression by (**a**) SDS-PAGE at 12%, stained with Coomassie blue; (**b**) Western blotting using anti-His tag Ab. M: MW marker, 1: Cell extract of *E. coli* BL21(DE3) transformed with pET28a, 2: supernatant after disruption of the cell extract of *E. coli* transformed with pS2NDH plasmid, 3: pellet after disruption of the cell extract of *E. coli* transformed with pS2NDH plasmid. (**c**) Analysis of the purified protein by SDS-PAGE at 12%, stained with Coomassie blue. Transmission electron microscopy of S2NDH protein (**d**) and S2NDH + ODN-39M (**e**): magnification 100,000× (bar equals 100 nm). (**f**) Recognition, by SARS-CoV-2 positive and negative human sera (1:2000 diluted), of S2NDH protein (left panel), S2NDH + ODN-39M (middle panel), and ODN-39M (right panel). Data are presented as OD_492 nm_ values from each individual serum. Horizontal bar represents the mean of the group in each case.

**Figure 2 vaccines-12-00588-f002:**
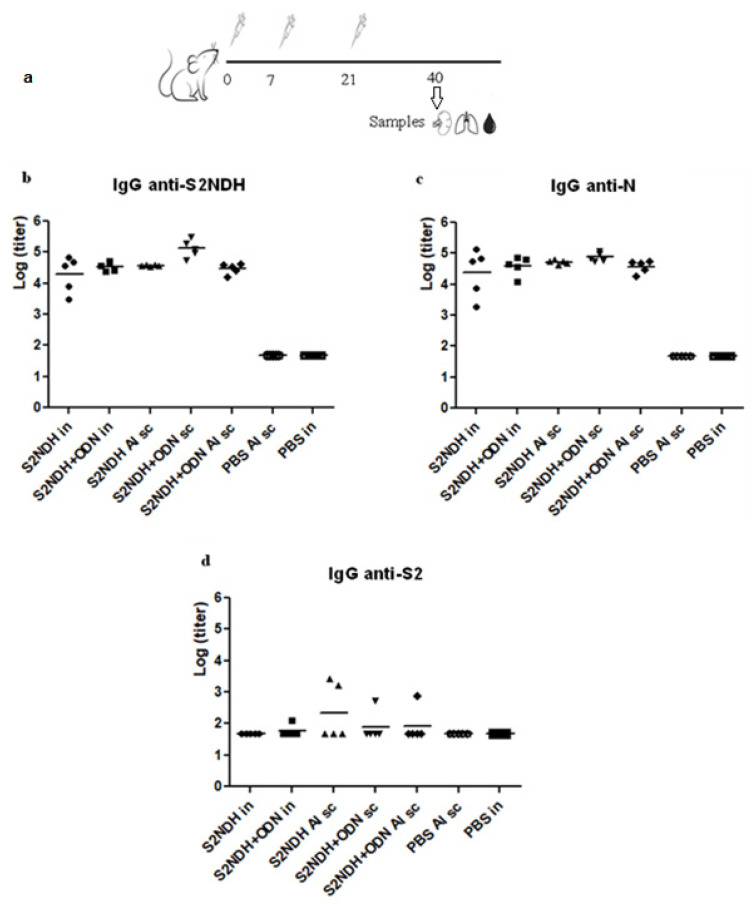
IgG antibody response, measured by ELISA, in sera from Balb/c mice immunized with S2NDH based formulations. (**a**) Study diagram. Mice were immunized with three doses (0, 7, 21 days) of each formulation by different routes, intranasal (in) or subcutaneous (sc). Nineteen days after the third dose, mice were sacrificed, and IgG responses were evaluated: (**b**) anti-S2NDH, (**c**) anti-N, and (**d**) anti-S2. Data are represented as log of the titers. The statistical analysis was carried out by one-way ANOVA followed by Tukey’s multiple comparison test.

**Figure 3 vaccines-12-00588-f003:**
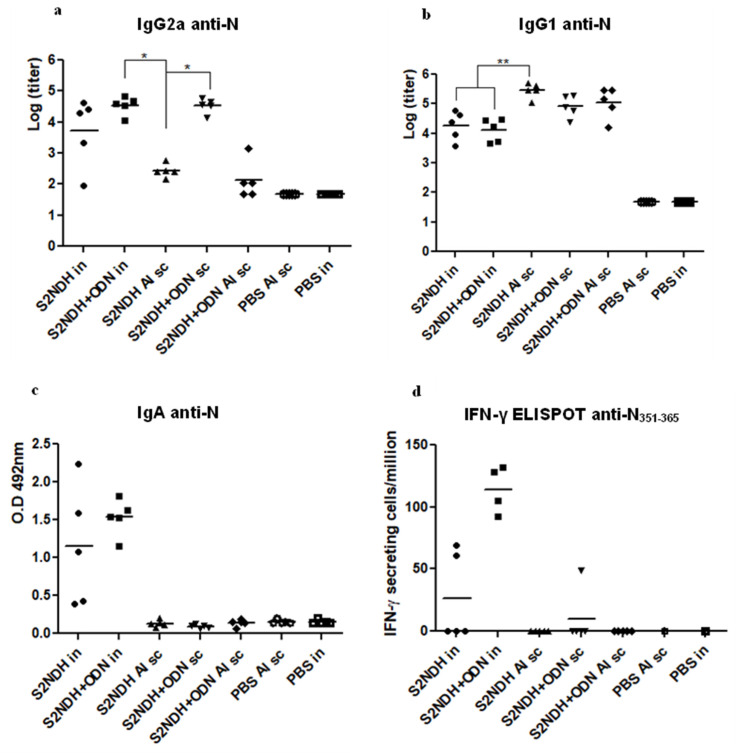
Immune response elicited by S2NDH formulations. Balb/c mice were immunized with three doses (0, 7, 21 days) of each formulation by different routes, intranasal (in) or subcutaneous (sc). Nineteen days after the third immunization, mice were sacrificed, and antibody responses were evaluated. Anti-N IgG subclasses measured in sera by ELISA (**a**) IgG2a and (**b**) IgG1. Data are represented as log of the titers. The statistical analysis was carried out by one-way ANOVA followed by Tukey’s multiple comparison test. * *p* < 0.05, ** *p* < 0.01 (**c**) Anti-N IgA antibody response measured in bronchoalveolar fluid (BALFs) by ELISA. Data are expressed as O.D 492 nm; the horizontal bar represents the mean. (**d**) Cell-mediated immune response. Nineteen days after the third dose, mice were sacrificed; the spleen cells were isolated and in vitro stimulated with the N_351–365_ peptide. The frequency of IFN-γ secreting cells was measured by ELISPOT. The horizontal bar represents the mean.

**Figure 4 vaccines-12-00588-f004:**
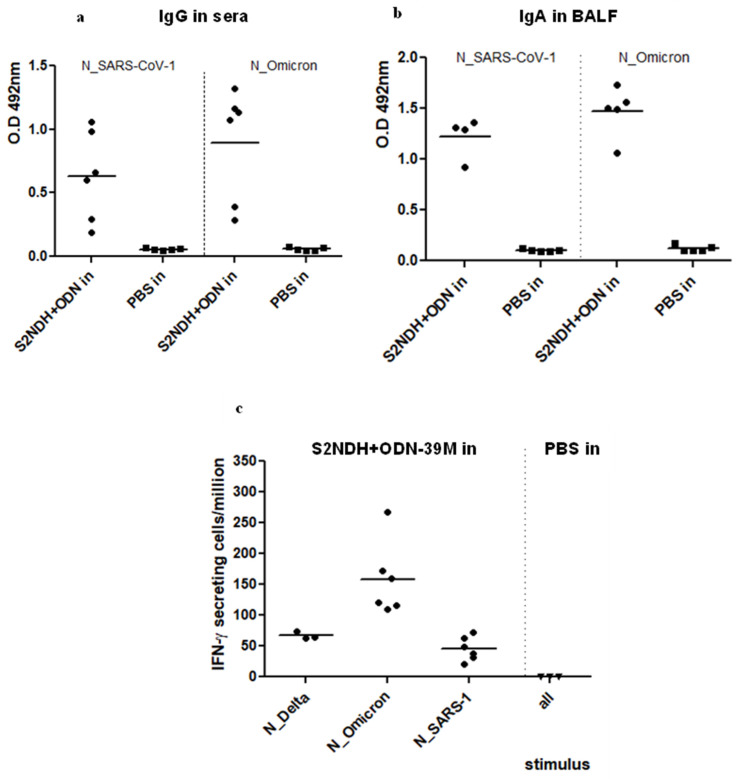
Cross-reactivity of the immune response generated by the intranasal administration of S2NDH + ODN-39M. Balb/c mice were intranasally immunized with three doses. After the third immunization, mice were sacrificed, and antibody responses against N protein from SARS-CoV-2 Omicron and from SARS-CoV-1 were measured by (**a**) IgG ELISA in sera, 1:1000 dilution and (**b**) IgA ELISA in bronchoalveolar fluid (BALF), without dilution. Data are expressed as the O.D 492 nm values. (**c**) Cell-mediated immune response. Three or six animals per group were sacrificed, and spleen cells were isolated and stimulated in vitro with N protein from SARS-CoV-2 Delta and Omicron variants and SARS-CoV-1. The frequency of IFN-γ secreting cells was measured by ELISPOT. The horizontal bar represents the mean.

**Figure 5 vaccines-12-00588-f005:**
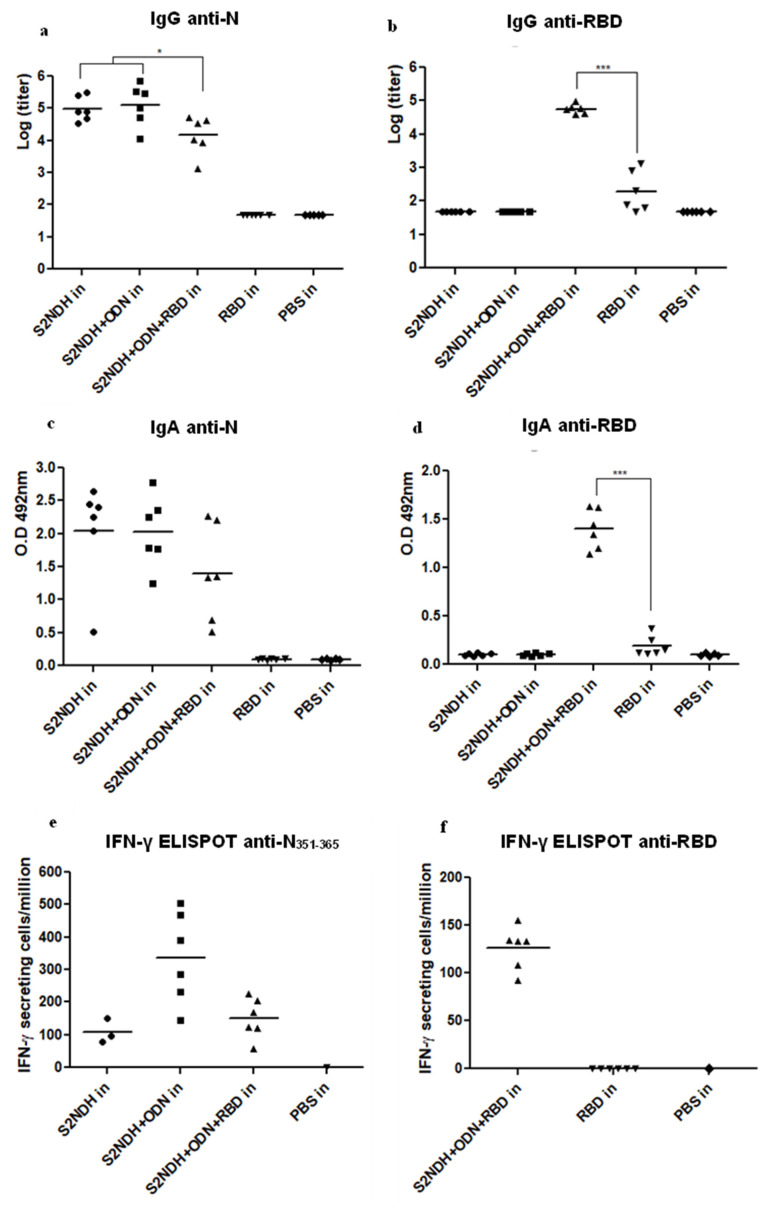
Immune response generated by intranasal administration of the bivalent formulation S2NDH + ODN-39M + RBD. Balb/c mice were immunized with three doses (0, 15, 30 days). Twenty-seven days after the third immunization, mice were sacrificed, and antibody response was evaluated by ELISA. IgG measured in sera (**a**) anti-N and (**b**) anti-RBD. Data are represented as log of the titers. IgA measured in BALFs (**c**) anti-N and (**d**) anti-RBD. Data are expressed as O.D 492 nm; the horizontal bar represents the mean. The statistical analysis was carried out by one-way ANOVA followed by Tukey’s multiple comparison test. * *p* < 0.05, *** *p* < 0.001. Cell-mediated immune response. At the same time point, spleen cells were isolated and in vitro stimulated with (**e**) N_351–365_ peptide and (**f**) RBD. The frequency of IFN-γ secreting cells was measured by ELISPOT. The horizontal bar represents the mean.

**Figure 6 vaccines-12-00588-f006:**
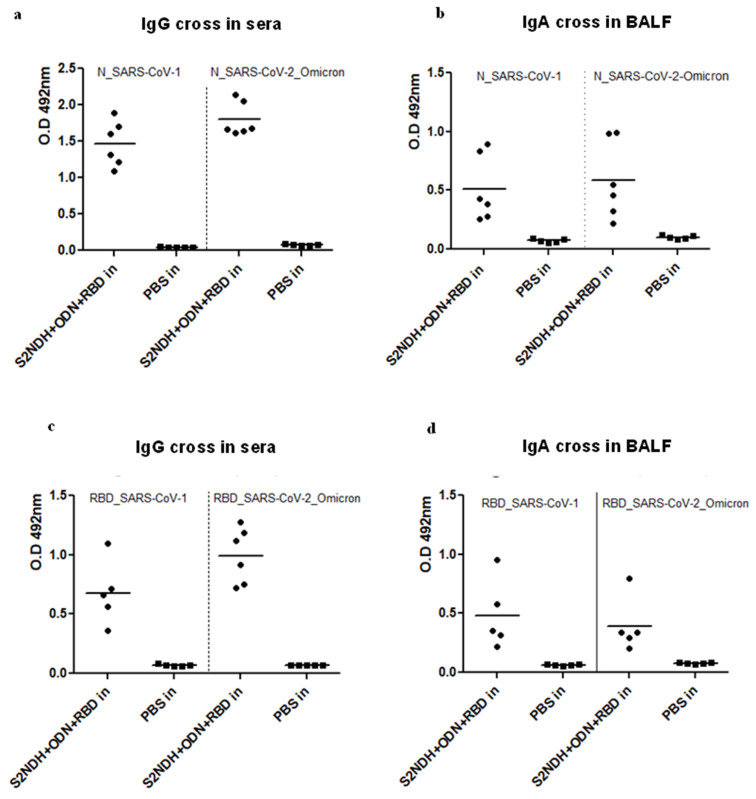
Cross-reactive antibody immune response generated after intranasal administration of the S2NDH + ODN + RBD bivalent formulation. After three doses, mice were sacrificed, and antibody responses were evaluated by ELISA. IgG measured in sera 1/100 dilution (**a**), and IgA measured in BALFs (without dilution) (**b**), against N protein from SARS-CoV-2 Omicron and SARS-CoV-1. IgG in sera (1:100 diluted) anti-RBD from SARS-CoV-2 Omicron and SARS-CoV-1 (**c**). IgA in BALF without dilution (**d**) against RBD from SARS-CoV-2 Omicron and SARS-CoV-1. Data are expressed as O.D 492 nm, the horizontal bar represents the mean.

**Figure 7 vaccines-12-00588-f007:**
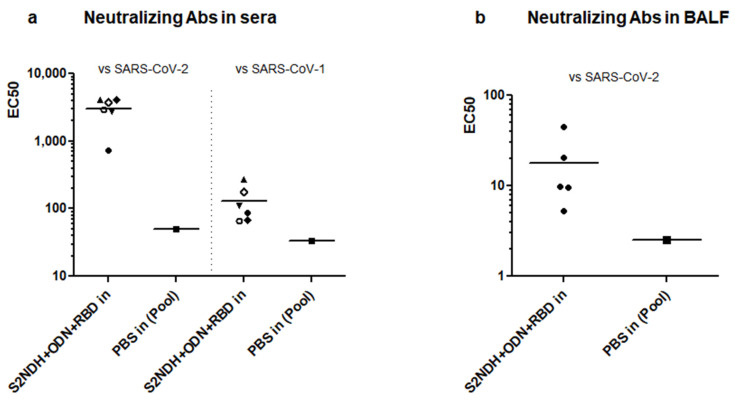
Neutralizing antibodies induced by the bivalent formulation S2NDH + ODN-39M + RBD intranasally administered (**a**) in sera (**b**) in BALFs. Neutralization tests were performed using pseudotyped VSV system, carrying the spike protein from the Ancestral variant of SARS-CoV-2 or SARS-CoV-1. The graphic represents the EC_50_ using a log10 scale, different symbols were used to identify each mouse, the bar represent the mean value.

## Data Availability

The original contributions presented in the study are included in the article/Appendix A, further inquiries can be directed to the corresponding author/s.

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
