# Peer review of "A Nasal Vaccine Candidate, Containing Three Antigenic Regions from SARS-CoV-2, to Induce a Broader Response"

_vaccines, 2024, doi:10.3390/vaccines12060588_

Round 1

Reviewer 1 Report

Comments and Suggestions for Authors

The article discusses the development of a bivalent subunit intranasal vaccine against SARS-CoV-2. The authors show that chimeric protein from the N and S proteins in addition to RBD protein in an adjuvanted formulation with CpG causes mucosal and systemic nAb generation against SARS-CoV-1 and SARS-CoV-2. 

Overall, the investigations are well presented and I have a few questions and points for the authors to address:

1. Primer information should be included in the methods for reproducibility of the work

2. Can the authors elaborate on the choice of a subcutaneous injection instead of an intramuscular injection to compare to intranasal? As the i.m. route is more relevant for subunit vaccines.

3. How was the 10ug dose selected?

4. In Section 3.1, the authors mention quantitative data with respect to Fig 1; e.g., "15% of the total cellular proteins," "more than 95% of purity.", etc. How are the authors arriving at these numbers? If quantitative methods are used, they should be described and reported. 

5. In Figure 1d, the the scale bar is not visible and must be added in order to visualize the particles. The authors claim that 10 nm sized particles were observed. The authors should strongly consider DLS or other techniques to properly characterize the hydrodynamic diameter of the particles as this is a liquid formulation. 

6. In Fig5, why was the vaccination schedule changed from 0, 7, 21 days to 0, 15, 30 days? This obviously has a major effect on the responses and must be discussed. 

7. In the discussion, authors should mention more clearly the differences in RBD and S protein structures across different across different SARS-CoV-2 variants and how the resultant responses against the variants can be explained.

8. Overall, statistical results should be more thoroughly included in results and discussion sections. The authors should point out when results are statistically insignificant by using "ns" as described in the methods. 

Comments on the Quality of English Language

Overall, the English Language of the manuscript is in very good shape. Only minor review is needed to correct typos and grammatical errors.

Author Response

Dear Reviewer,

Thank you very much for your valuable revision. You will find in the corrected manuscript (with track changes) the modifications according to the similarity report. In turn, the modifications based on the reviewer’s revision, are indicated underlined in bold.

Yours sincerely,

Dr. Lisset Hermida

Reviewer 2 Report

Comments and Suggestions for Authors

The manuscript by Lobaina et al., described the design and immunogenicity study of a nasal vaccine candidate containing three antigenic regions of SARS-CoV-2. The authors conducted multiple experiments to evaluate the vaccine by comparing different adjuvants and the combinatory use of RBD subunit protein. However, there are some concerns that need to be addressed.

Major concerns:

1. It appears that the authors should exert more effort to highlight the novelty of designing such a chimeric antigen. Additionally, the humoral responses elicited by the vaccine candidates were only assessed at one time point (e.g., 19 days after the final boost immunization in Figure 2), leaving the kinetics of antigen-specific antibody response unknown. Hence, it is difficult to judge which candidate performs best based solely on the current data presented.

2. It is well-known that the T-cell response elicited by vaccines peaks within 7-14 days upon vaccination. However, in this study, the authors evaluated the cellular response too late (day 40), missing the appropriate timing to compare vaccine-mediated T-cell responses among groups.

3. While it is shown that the combinatory vaccination of S2NDH + ODN + RBD was able to induce neutralizing antibodies against SARS-CoV-2 and SARS-CoV-1, it is recommended that a challenge experiment in animals (at least for the lead group versus the control group) should be conducted to confirm the protective potency against viral challenge.

Minor remarks:

1. Lines 133-135 are similar to the texts of Lines 139-141; however, the authors stated them differently: “continue culturing for 4 hours at 37°C” (Line 135) vs. “for inducing gene expression overnight at 37°C” (Line 142). The same applies to the different centrifuge speeds mentioned (Line 136 vs. Line 143).

2. Several typos were found in the current manuscript (e.g., “40C” “uL” shoule be corrected to “4°C” “μL”).

3. In Line 183, a dilution of 1/300 for the goat anti-rabbit IgG secondary antibody seems abnormal.

4. In Line 214, “Six to 8 weeks old” should be consistently stated in the same form.

5. In Line 273, “a 96 well culture plate” should be used instead of “a 96-well culture plate”.

6. In the results section, sections 3.1-3.3 could be better combined into one part, as their results are depicted in Figure 1.

Author Response

(The authors gave the same response as above.)

Round 2

Reviewer 2 Report

Comments and Suggestions for Authors

Authors should carefully check typos and maintain consistency in, for example, units and texts in the manuscript.

Author Response

Dear Reviewer, we really appreciate your further revision. Following your recommendations we carefully re-checked and corrected the typos mistakes and eliminated any inconsistency in the nomenclature or units in the body of the manuscript.

Best Regards,